# Semi-Automatic Corpus Expansion and Extraction of Uyghur-Named Entities and Relations Based on a Hybrid Method

**Ayiguli Halike** [1,2], **Kahaerjiang Abiderexiti** [1,2] **and Tuergen Yibulayin** [1,2,*]

[1] School of Information Science and Engineering, Xinjiang University, Urumqi 830046, Xinjiang, China; ayiguli@stu.xju.edu.cn (A.H.); kaharjan@xju.edu.cn (K.A.)

[2] Multilingual Information Technology Laboratory of Xinjiang University, Urumqi 830046, Xinjiang, China

[*] Correspondence: turgun@xju.edu.cn

**Abstract:** Relation extraction is an important task with many applications in natural language processing, such as structured knowledge extraction, knowledge graph construction, and automatic question answering system construction. However, relatively little past work has focused on the construction of the corpus and extraction of Uyghur-named entity relations, resulting in a very limited availability of relation extraction research and a deficiency of annotated relation data. This issue is addressed in the present article by proposing a hybrid Uyghur-named entity relation extraction method that combines a conditional random field model for making suggestions regarding annotation based on extracted relations with a set of rules applied by human annotators to rapidly increase the size of the Uyghur corpus. We integrate our relation extraction method into an existing annotation tool, and, with the help of human correction, we implement Uyghur relation extraction and expand the existing corpus. The effectiveness of our proposed approach is demonstrated based on experimental results by using an existing Uyghur corpus, and our method achieves a maximum weighted average between precision and recall of 61.34%. The method we proposed achieves state-of-the-art results on entity and relation extraction tasks in Uyghur.

**Keywords:** relation extraction; named entity; hybrid neural network; conditional random field; Uyghur

---

## 1. Introduction

Extracting entities and relations from unstructured texts is crucial for knowledge base construction in natural language processing (NLP) [1–4], intelligent question answering systems [5,6], and search engines. The development of knowledge graphs is particularly well-suited to this purpose due to their well-structured nature that typically comes in the form of three entries that are denoted as a head entity, a relation, and a tail entity (or h, r, and t). In particular, the automatic construction of knowledge graphs based on unstructured data has attracted significant interest. For example, projects such as DBPedia [7], YAGO [8], Kylin/KOG [9,10], and BabelNet [11] have focused on building knowledge graphs by using entities and relations that are extracted from unstructured data obtained from Wikipedia, which is one of the largest sources of multilingual language data on the internet. However, relation extraction requires sufficient annotated corpus data, particularly in supervised learning. Nonetheless, supervised relation extraction models [12,13] usually suffer from a lack of high-quality training data because the manual labeling of data is human labor-intensive and time-consuming. The above-discussed issue is particularly problematic for Uyghur because relatively little past work has focused on the construction of the corpus and has not included research about the extraction of Uyghur-named entity relations, resulting in the very limited availability of relation extraction research and a deficiency of

annotated relation data. This lack of sufficient annotated corpus data is a significant challenge that has affected efforts toward the extraction of entities and relations from unstructured Uyghur texts. Moreover, relatively few annotated named Uyghur entity relation data are available on the internet. In addition, we note that the construction of knowledge graphs is quite language-dependent [14]. Currently, most knowledge graph systems are developed in the English language and then directly translated into the language of interest. However, this approach is often not feasible for Uyghur because the direct translation of English knowledge graphs into Uyghur is not always possible. As such, knowledge graphs must be directly constructed in Uyghur based on extracted relations to develop a reasonably sophisticated Uyghur knowledge base. However, Uyghur relation extraction is quite difficult compared to that of the English language because the Uyghur language is morphologically complex, and the difficulty of the task is further compounded by the limited availability of annotated data. These issues represent major limitations that affect the extraction of entities and relations in the Uyghur language. Uyghur is a type of morphologically-rich agglutinative language that is used by approximately 10 million people in the Xinjiang Uyghur autonomous region of China. Uyghur words are formed by a root followed by suffixes [14,15]; therefore, the size of the vocabulary is huge. Officially, Arabic and Latin scripts are used in Uyghur, while Latin scripts are also widely used on social networks and mobile communications.

The present work seeks to alleviate the above-discussed limitations by proposing a hybrid neural network and semi-automatic named-entity relation recognition method for making suggestions regarding annotation based on extracted relations with a set of rules applied by human annotators to expand the existing annotated Uyghur corpus more rapidly than by human annotation alone. We also focus on the issues raised during Uyghur knowledge graph construction and discuss the main challenges that must be addressed in this task. Finally, the effectiveness of our proposed approach is experimentally demonstrated.

## 2. Related Works

Entity relations are the key components that are required for building knowledge graphs, and numerous methods have been developed for relation recognition and extraction [16]. Conventional approaches consider entity recognition to be an antecedent step in a pipeline for relation extraction [17–19]. However, the dependence between the two tasks is typically ignored. The relation extraction task can be seen as a sequence labeling problem. As such, numerous methods have been based on sequence tagging. For example, research has been conducted to develop an entity relation descriptor based on a linear-chain Conditional Random Field (CRF) model, which has been demonstrated to reduce the space of possible label sequences and introduce long-range features [20–23]. Numerous sequence tagging approaches have been developed, including hidden Markov models (HMM), maximum entropy Markov models [10], CRF models [24], and neural network methods [25–29]. Among these, CRF models have been a commonly used in Natural Language Processing (NLP) applications in recent years. A CRF model combines maximum entropy with a hidden Markov model, which is a typical non-directional pattern model of discriminant probability. A CRF attempts to model the conditional probability of multiple variables after observing a given value. As such, the construction of a conditional probability model is the goal of a CRF [26]. Neural network-based methods have also become widely used in NLP applications. For example, the authors of [30] proposed a multichannel convolutional neural network (MCCNN) for automated biomedical relation extraction that obtained an average accuracy of 90.1%. However, the accuracy of named entity recognition affects the accuracy of relation extraction. This has recently been addressed by the joint extraction of named entities and relations [30–32]. For example, the authors of [33] proposed a novel tagging scheme and demonstrated that the sentence annotations could be applied to the joint extraction task based on different end-to-end models. The authors of [34] proposed a globally optimized neural model, and they achieved the best relation extraction performance for existing state-of-the-art methods on two standard benchmarks. The authors of [35] proposed attention-based bidirectional long short-term memory (BILSTM) with a

conditional random field layer for document-level chemical Named Entity Recognition (NER). For NLP applications specific to the Uyghur language, a few studies have focused on corpus construction. Here, the authors of [14] proposed a method for constructing a Uyghur-named entity and relation corpus to expand the size of the existing corpus. The authors of [36] constructed a contemporary Uyghur grammatical information dictionary that provided extensive grammatical information and collocation features and is presently a primary resource for NLP research specific to the Uyghur language. The Uyghur Dependency Treebank was built from a public reading corpus [37,38], and also presently serves as an important tool for Uyghur linguistic studies. A standard scheme for tagging Uyghur sentences to construct a typical knowledge graph is illustrated in Table 1.

**Table 1.** A standard Uyghur sentence tagging scheme.

| Type | Content (in Uyghur) | Content (in English) |
|---|---|---|
| Sentence | Alim Adilning ayali Aliye Ubul. | Alim Adil's wife is Aliye Ubul |
| First entity | Alim Adilning | Alim Adil's |
| Relation type | Personal.Family | Personal.Family |
| Tail entity | Aliye Ubul | Aliye Ubul |

For convenience, all the examples in the articles are shown in Uyghur Latin script form. Our experiment data are in Arabic script form.

It can be seen from Table 1 that "Alim Adilning" and "Aliye Ubul" are the first and second entities in the sentence, respectively. "Personal.Family" is a relation type.

## 3. Uyghur Relation Extraction Model

The present study adopts a hybrid neural network and CRF model to analyze the grammatical and semantic features of Uyghur-named entity relations. Experimental results have shown that a CRF model provides better extraction performance than neural network models when the amount of data is relatively small.

### 3.1. Task Definition

As mentioned above, the relation extraction task can be seen as a sequence labeling problem, and all Uyghur datasets applied to the hybrid neural network and CRF model must be labeled. The annotation for relation extraction from raw texts consists solely of relation extraction tags that recognize valid relations between entity pairs. The present work constructed feature sets of the different entity categories based on the characteristics of Uyghur-named entities and relations. These features include word-related features and dictionary features. Word-related features include Uyghur words, part-of-speech tagging, syllables, word lengths, and syllable lengths. We neglected the use of word stem characteristics because Uyghur word stemming is complicated, and no reasonably good stemming tools are presently available for Uyghur. Meanwhile, numerous dictionary features were adopted, such as common dictionaries, person name dictionaries, place name dictionaries, organization name dictionaries, and similarity dictionaries based on word vectors. Table 2 presents the annotation that were employed for each word in an example Uyghur sentence to facilitate relation extraction. Each sentence has a number of tags, where "O" represents "other" (which is independent of the extracted results), "S" represents a single word, "B" represents the first word of an entity, "I" represents the one middle word of the entity, and "E" represents the last word of the entity. The annotations adopted for an example Uyghur sentence to facilitate relation extraction are shown in Table 2.

**Table 2.** Annotations adopted for an example Uyghur sentence to facilitate relation extraction.

| Uyghur | English | Tag |
|--------|---------|-----|
| Xintian | Xintian | B_Org–Aff.Employment_2 |
| shirketi | company | I_Org–Aff.Employment_2 |
| kurghuchisi | founder | E_Org–Aff.Employment_2 |
| Zilu | Zilu | S_Org–Aff.Employment_1 |
| xongliyen | Honglian | B_Org–Aff.Employment_4 |
| shirkiti | company | I_Org–Aff.Employment_4 |
| kurghuchisi | founder | E_Org–Aff.Employment_4 |
| dupëng | Du peng | B_Org–Aff.Employment_3 |
| bilen | and | O |
| körüxti | see | O |
| . | . | O |

In the above sentence, tags "O," "S," "B," "I," and "E" represent other, a single word, the first word of an entity, the one middle word of the entity, and the last word of the entity, respectively, and the middle term "Org–Aff.Employment" represents the relation between two entities where "Org–Aff" is a subtype of "Employment" and the terminating number represents the entity to which the word belongs (e.g., "Zilu" is the first entity).

### 3.2. Feature Template

The effect of combining different features on the named entity relations extraction process cannot be ignored. Therefore, the selection of feature templates plays an extremely important role in relation extraction. The processes of named entity relations recognition and extraction must both consider individual words and the context of each word (i.e., its surrounding words), and the CRF model must be designed to synthesize contextual information as well as external features. In this paper, we used CRF Sharp open source tools to build the CRF model for conducting Uyghur-named entity relation recognition, and a supervised corpus was employed to predict the relation type based on the CRF model. The template features adopted in the current work are listed in Table 3. Here, we adopted not only an atomic feature (unary feature) template but also a composite feature template that represents a combination of three features, while the other three features represents binary feature combinations. In addition, $w$ represents the first column of the corpus, which is a column of words, and F denotes other characteristic columns without words. In addition to the feature categories, the size of the feature window must also be considered when establishing a CRF model because the window contains the contextual information of a word. An overly large window will lead to information redundancy and reduce the training thickness of the model. Meanwhile, an overly small window size will provide insufficient information for model training, and the extraction performance of the model will suffer. In this paper, the final selected window size was 4 + 1.

**Table 3.** The definitions of feature templates.

| Feature Type | Template | Meaning |
|--------------|----------|---------|
| Atomic feature | $w_i\ (-2 \le i \le 2, i \in z)$ | Current word $w_i$ and the words of its upper and lower two windows; the window size is 5. |
|  | $F_i\ (-1 \le i \le 1, i \in z)$ | Characteristics of the current word $F_i$ and the words of its upper and lower windows; the window size is 3. |
| Composite feature | $w_{i-1}\big|w_i(0 \le i \le 1, i \in z)$ | Combination of the current word and the word in its upper window. |
|  | $F_{i-1}\big|F_i(0 \le i \le 1, i \in z)$ | Combination feature of the current word and the word in its upper window. |
|  | $w_i\big|F_i(i = 0)$ | Current word and its combination features. |
|  | $F_{i-1}|F_i|F_{i+1}\ (i = 0)$ | Characteristics of the current word and the combination features of its upper and lower windows. |

All sentences are treated as sequences in a CRF model, where each word in a sentence is a moment. In the process of relationship identification, a CRF model calculates the probability of applying particular tags to the moments of a sequence based on the acquired features and weights of each sequence, and these probabilities are then used as the input parameters of the conditional probability. The CRF model then normalizes the distribution of the entire state sequence. Furthermore,

the selection of feature sets directly affects the performance of the model. Therefore, we built a candidate feature set of useful features to determine which feature had the strongest efficacy for predicting the entity relations.

The rows of a corpus selected by a feature are identified relative to a given position, while the selected columns of the corpus are identified according to an absolute position. Generally, m rows before and after a given row are selected, and n − 1 (where n is the total number of columns in the corpus) columns are selected. Each line in the template file is a feature template. The feature templates are represented as a token in the input data according to the statement %x[row, column], where row specifies the row offset to the current token and column specifies the position of the selected column. The initial position is 0. Unigram features have the letter U affixed to them, and bigram features are affixed with the letter B, as shown in Table 4.

**Table 4.** Feature template table for example words.

| Feature Template | Feature Meaning | Representative Character | English |
|---|---|---|---|
| %x[−2, 0] | −2 rows from the current row, column 0 | Alim | Alim |
| %x[−1, 0] | −1 row from the current row, column 0 | adilning | Adil's |
| %x[0, 0] | 0 rows from the current row, column 0 | ayali | wife |
| %x[1, 0] | 1 row from the current row, column 0 | Aliye | Aliye |
| %x[2, 0] | 2 rows from the current row, column 0 | Ubul | Ubul |
| %x[3, 0] | 3 rows from the current row, column 0 | . | . |
| %x[−1, 0]/%x[0, 0] | −1 row from the current row, column 0, combination of row 0 and column 0 | adilning/ayali | Adil's/wife |
| %x[0, 0]/%x[1, 0] | row 0, column 0, the combination of row 1 and column 0 | ayali/Aliye | wife/Aliye |

Here, when a template is denoted as "U01:%x[0, 1]", the CRF model produces the feature function sets func1, . . . , funcN.

### 3.3. Rules

The relation is different from the named entity. The task of Uyghur relation extraction has been simplified in the present work by assuming that the named entity was given. Therefore, only the relationships between the named entities represented in a sentence were tagged. However, the results obtained in this manner were not particularly useful for the subsequent task of human annotation. Thus, rules were adopted to denote the relation type after conducting relation extraction based on the CRF model. These rules were denoted according to the standard labeling shown in Table 5. Here, the rules for physical location relations were that the first parameter of this relationship had to be a person (PER), while the second parameter could be a facility (FAC), location (LOC), or a geographical, social, or political entity (GPE).

**Table 5.** Input format of entity relationships.

| Types | Subtypes |
|---|---|
| Part–Whole | Part–Whole.Geo |
| | Part–Whole.Subsidiary |
| Per–Social | Per–Social.Business |
| | Per–Social.Family |
| | Per–Social.Role |
| | Per–Social.Other |
| Physical | Physical.Located |
| | Physical.Near |
| Org–Aff | Org–Aff.Employment |
| | Org–Aff.Investor–Shareholder |
| | Org–Aff.Student–Alum |
| | Org–Aff.Owner |
| | Org–Aff.Founder |
| Gen–Aff | Gen–Aff.Person–Age |
| | Gen–Aff.Organizationwebsite |

As an example, the permissible parameter table for Physical.Located is given in Table 6 for the sentence "adil shangxeyde oquwatidu" (Adil is studying at Shanghai). These permissible parameters are shown in Table 6.

**Table 6.** Permissible parameter table for an example sentence.

| Relation Type | Arg1 | Arg2 |
|---|---|---|
| Physical.Located | PER | FAC, LOC, GPE |
| Entity | adil | shangxeyde |
| English | Adil | At the Shanghai |

Here, "shangxeyde" is Arg2, which is an LOC. The rules of other types are similar to the presented example.

### 3.4. Hybrid Neural Network Model Training

Hybrid neural network models are based on convolutional neural networks (CNN) and long short-term memory (LSTM). In this paper, we propose a CRF model and hybrid neural network based on the semantic and morphological features in Uyghur. Both methods have their advantages and disadvantages.

The framework of the hybrid neural network is shown in Figure 1. The first layer is a bidirectional LSTM-encoding layer that is shared by the relation extraction and named entity extraction models. There are two modules after the encoding layer. One is the named entity recognition module and is linked to an LSTM-decoding layer, and the other feeds into a CNN layer to extract the relations. The basic layer contains word embedding, relation, the position of the entity and arg_type.

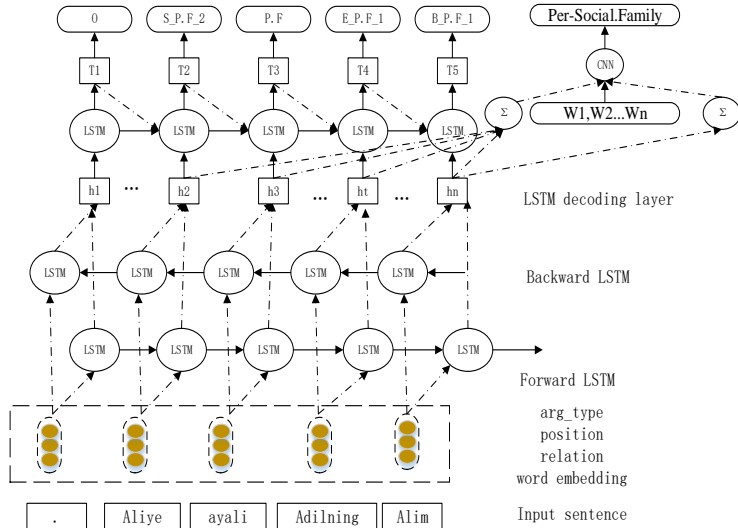

**Figure 1.** The framework of the hybrid neural network for jointly extracting entities and relations.

### 3.4.1. Bidirectional LSTM-Encoding Layer

The bidirectional LSTM-encoding layer [39] contains a word embedding layer, a forward LSTM layer, a backward LSTM layer, and a concatenation layer. The word embedding layer converts the word with 1-hot representation to an embedding vector. A sequence of words is represented as $W = \{w_1, \dots w_t, w_{t+1}, w_n\}$, where $w_t \in R^d$ is the dimensional word vector corresponding to the t-th word in the sentence with a length equal to the input sentence. After the word embedding layer, there are two parallel LSTM layers: the forward LSTM layer and the backward LSTM layer.

We express the detailed operation of LSTM in Equations (1)–(6). $i_t$ is the input gate that controls how much of the current input $x_t$ and previous output c will enter into the new cell. $f_t$ decides whether

to erase or keep individual components of the memory. For each word $w_t$, the forward layer $h_t$ encodes $w_t$ by considering the contextual information from words $w_1$ to $w_t$.

$$i_t = \delta(W_{wi}w_t + W_{hi}h_{t-1} + W_{ci}c_{t-1} + b_i) \tag{1}$$

$$f = \delta\left(W_{wf}w_t + W_{hf}h_{t-1} + W_{cf}c_{t-1} + b_f\right) \tag{2}$$

$$z_t = tanh(W_{wc}w_t + W_{hc}h_{t-1} + b_c). \tag{3}$$

$$c_t = f_t c_{t-1} + i_t z_t \tag{4}$$

$$o_t = \delta(W_{wo}w_t + W_{ho}h_{t-1} + W_{co}c_t + b_o) \tag{5}$$

$$h_t = o_t tanh(c_t) \tag{6}$$

In Equations (5)–(10), i, $f$, and $o_t$ are the input gate, forget gate, and output gate, respectively. b is the bias term, c is the cell memory, and $W$ represents the parameters. We concatenate $h_t$ to represent word t's encoded information.

### 3.4.2. LSTM-Encoding Layer

We used an LSTM structure to produce the tag sequence, as given in Equations (11)–(19). When detecting the tag of word $w_t$, the inputs of the decoding layer consist of $h_t$ obtained from the bidirectional LSTM layer, the former predicted tag embedding $T_{t-1}$, the former cell value $c_{t-1}^{(2)}$, and the former hidden vector in decoding layer $h_{t-1}^{(2)}$.

$$i_t^{(2)} = \delta\left(W_{wi}^{(2)}h_t + W_{hi}^{(2)}h_{t-1}^{(2)} + W_{ti}T_{t-1} + b_i^{(2)}\right) \tag{7}$$

$$f_t^{(2)} = \delta\left(W_{wf}^{(2)}h_t + W_{hf}^{(2)}h_{t-1}^{(2)} + W_{tf}T_{t-1} + b_f^{(2)}\right) \tag{8}$$

$$z_t^{(2)} = tanh(W_{wc}^{(2)}h_t + W_{hc}^{(2)}h_{t-1}^{(2)} + W_{tc}T_{t-1} + b_c^{(2)} \tag{9}$$

$$c_t^{(2)} = f_t^{(2)}c_{t-2}^{(2)} + i_t^{(2)}z_t^{(2)} \tag{10}$$

$$o_t^{(2)} = \delta\left(W_{wo}^{(2)}w_t^{(2)} + W_{ho}^{(2)}h_{t-1}^{(2)} + W_{co}^{(2)}c_t^{(2)} + b_o^{(2)}\right) \tag{11}$$

$$h_t^{(2)} = o_t^{(2)}tanh\left(c_t^{(2)}\right) \tag{12}$$

$$T_t = W_{ts}h_t^{(2)} + b_{ts} \tag{13}$$

$$y_t = W_y T_t + b_y \tag{14}$$

$$p_t^i = \frac{exp(p_i)}{\sum_{j=1}^{N_t} exp\left(y_t^j\right)} \tag{15}$$

In Equations (7)–(15), $w_y$ is the softmax matrix and $N_t$ is the total number of tags. The tag prediction vector is similar to tag embedding, and LSTM is capable of learning long-term dependencies. Thus, the decoding method can model tag interactions.

### 3.4.3. LSTM-Decoding Layer

When extracting entities' semantic relations, we merge the encoding information of entities and then feed them into the CNN model.1.

$$Relation\_type = CNN(h_{e_1}, w_{e_1}, \ldots w_{e_2}) \tag{16}$$

where $h_{e_1}$ is the encoding information of entity and $w_{e_1}$ is word embedding. The CNN denotes the convolutional operations. In the convolutional layer, $w_c^{(i)}$ represents the $i$-th convolutional filter, $br^{(i)}$ represents the bias term, and filter $w_c^{(i)}$ slides through the input sequence to get the features $z^i$. The sliding process can be represented as follows:

$$z_l^i = \sigma\left(w_c^{(i)} * s_{l:l+k-1} + br^{(i)}\right) \tag{17}$$

We apply the max-pooling operation to reserve the most prominent feature of filter $w_c^{(i)}$ and denote it as:

$$z_{max}^{(i)} = \max\left\{z^{(i)}\right\} = \max\left\{z_1^{(i)}, \dots z_{l-k+1}^{(i)}\right\} \tag{18}$$

Finally, we set the Softmax layer with dropout to classify the relations based on relation features $R_s$, which is defined as follows:

$$y_r = w_R \bullet (R_S \,^{\circ}\, r) + b_R \tag{19}$$

$$p_r^{(i)} = \frac{\exp\left(y_r^i\right)}{\sum_{j=1}^{nc} \exp\left(y_r^j\right)} \tag{20}$$

Here, $W_R \in R^{nr:nc}$ is the Softmax matrix, nc is the total number of relation classes, the symbol $^{\circ}$ denotes the element-wise multiplication operator, and $r \in R^{nr}$ is a binary mask vector that is drawn from b with probability p.

### 3.5. CRF Model

A CRF model is defined as follows.

Set s as a node of V and E as an undirected graph of a set with no edges.

Each node in V corresponds to a random variable whose value range is a possible set of tags.

Each random variable satisfies Markov characteristics for an observed sequence $X = \{x_1, x_2, x_3 \dots x_n\}$ for conditions as follows:

$$P(Y_V \backslash X, w \neq v) = p(Y_V \backslash X, Y_w, w \sim v) \tag{21}$$

where W and V denote two adjacent nodes in graph G. Then, (X, Y) is a CRF.

The CRF model calculates the conditional probability of an output node value under the condition of given input nodes. For a chain with corresponding weights $\theta = \theta_1, \theta_2, \dots, \theta_k$, the conditional probability of a state sequence obtained for a given sequence x is defined as follows:

$$P_\theta(Y|X) = \frac{1}{Z_x}\left\{\sum_{n=1}^{N}\sum_{k} f(y_{n-1}, y_n, X, n)\right\} \tag{22}$$

Here, the denominator $Z_x$ is a normalization factor that ensures that the sum of the probabilities of all possible state sequences is 1 for the given input $f_k(y_{n-1}, y_n, X, n)$, and that this holds for all values of $X$. Additionally, the feature functions, which are located at n and n − 1, may be 0, 1, or any real number. In most cases, the characteristic function is a binary representation function, where the value is 1 when the characteristic condition is satisfied; otherwise, it is 0. Accordingly, the characteristic functions are defined as follows:

$$f(y_i, x, i) = \begin{cases} 1 & \text{if } y_{i-1} = y \text{ and } y_i = y \\ 0 & \text{other} \end{cases} \tag{23}$$

$$g(y_i, x, i) = \begin{cases} 1 & \text{if } x_{i-1} = x \text{ and } y_i = y \\ 0 & \text{other} \end{cases} \tag{24}$$

The final step is to search for the value $Y* = \text{argmax } P(X\backslash Y)$, which represents the output with the highest probability.

Parameter settings: We used the CRF++ based NER as the baseline. The features used in CRF++ were unigrams and bigrams. The window size was set to 5, such that we considered two words ahead of and two words following the current word $w_t$.

## 4. Experiments

The raw statistics pertaining to the initial annotated Uyghur language corpus and expanded corpus are listed in Table 7. Reference [14] was focused on the construction of the corpus of Uyghur-named entity relations, resulting in the very limited availability of relation extraction research and a deficiency of annotated relation data. There are 571 documents, 6173 sentences, and 4098 useful sentences.

**Table 7.** Differences between original corpus and expanded corpus.

| Type | Original Corpus [14] | Expanded Corpus |
|---|---|---|
| Documents | 571 | 1032 |
| Filtered documents (relational documents) | 422 | 842 |
| Sentences | 6173 | 17,765 |
| Words | 27,846 | 1,142,241 |

Problems:

1. The corpus size was small.
2. There were also many sentences that did not have any entities and relations.
3. In this paper, the most basic corpus of Uyghur named entities and relations was constructed, and no study of the relation extraction research in Uyghur was undertaken.

Here, the 9103 sentences listed represent filtered sentences that were obtained from a total number of 17,765 sentences. This can address the first and second problems in the research of [14].

### 4.1. Dataset

As compared to the existing original corpus [14], our work expands the existing corpus size and improves its quality.

According to the conventions adopted in this work, an entity should be the name of a person (PER), location (LOC), organization (ORG), geographical entity (GPE), title (TTL), age (AGE), Uniform Resource Locater (URL), or facility (FAC).

The statistics of entities and their pertinence to these individual entity types are listed in Tables 8 and 9, respectively. We note from Table 9 that the number of entities is extremely unbalanced, in that the number of GPEs is very large while the number of URLs is very small. In addition, the total number of non-duplicated named entities in the annotated corpus is 5610.

The statistics pertaining to relations for the total number of 9103 filtered sentences in the initial annotated corpus are listed in Table 10.

The actual numbers of individual relation types observed in the initial annotated corpus are listed in Table 11. An analysis of Table 11 indicates that the corpus includes five relation types, which are listed in Table 12, and 16 subtypes, which are listed in Table 13.

**Table 8.** Entity statistics for the initial annotated corpus.

| Total Number of Entity Types | Entity Coverage (%) |
|---|---|
| 217,605 | 88.4 |

**Table 9.** Coverage of entity types in the initial annotated corpus.

| Entity Type | Entity Coverage (%) |
|---|---|
| GPE | 44.51 |
| ORG | 21.33 |
| PER | 16.71 |
| TTL | 8.46 |
| LOC | 7.21 |
| FAC | 1.41 |
| AGE | 0.29 |
| URL | 0.07 |

**Table 10.** Overall relation statistics for the initial annotated corpus.

| Number of Relations | Relation Coverage (%) |
|---|---|
| 4307 | 51.60 |

**Table 11.** Numbers of individual relation types observed in the initial corpus.

| ID | Relation Type | NUM | ID | Relation Type | NUM |
|---|---|---|---|---|---|
| 1 | Org–Aff.Employment | 534 | 9 | Physical.Near | 725 |
| 2 | Org–Aff.Owner | 234 | 10 | Gen–Aff.Organizationwebsite | 556 |
| 3 | Per–Social.Family | 240 | 11 | Per–Social.Other | 525 |
| 4 | Per–Social.Role | 1437 | 12 | Gen–Aff.Person–Age | 167 |
| 5 | Part–Whole.Geo | 513 | 13 | Org–Aff.Student–Alum | 154 |
| 6 | Org–Aff.Founder | 454 | 14 | Part–Whole.Subsidiary | 1475 |
| 7 | Org–Aff.Investor–Shareholder | 634 | 15 | Per–Social.Business | 168 |
| 8 | Physical.Located | 159 | | | |

**Table 12.** Relation type statistics for the initial corpus.

| Type | Percentage of Type (%) | Coverage of Type (%) |
|---|---|---|
| Physical | 4.32 | 2.23 |
| Part–Whole | 46.83 | 24.16 |
| Gen–Aff | 0.49 | 0.25 |
| Per–Social | 35.52 | 18.33 |
| Org–Aff | 12.84 | 6.63 |

**Table 13.** Relation subtype statistics for the initial corpus.

| Relation Subtype | Types (%) | Subtypes (%) |
|---|---|---|
| Subsidiary | 34.53 | 17.8 |
| Role | 33.83 | 17.4 |
| Employment | 12.54 | 6.47 |
| Per–Social | 35.52 | 18.33 |
| Org–Aff | 12.84 | 6.63 |
| Located | 3.74 | 41.93 |
| Other | 1.25 | 65.65 |
| Near | 45.58 | 45.30 |
| Family | 32.44 | 65.23 |
| Person–Age | 45.37 | 54.19 |
| Organization Website | 43.12 | 76.06 |
| Business | 32.00 | 43.00 |
| Investor–Shareholder | 54.14 | 54.07 |
| Student–Alum | 43.02 | 65.01 |
| Ownership | 67.05 | 54.02 |
| Founder | 54.09 | 65.05 |

An analysis of the tables indicates that most of the relation types are subsidiary. We also note from Table 13 that that the Part–Whole relation type includes the highest number of named entity relations, while the Gen–Aff relation type has the least.

*4.2. Experimental Results*

We first employed our proposed CRF model to conduct Uyghur relation extraction for 842 documents by using only word features *Fwc*. The performance results are listed in Table 14.

**Table 14.** Relation extraction performance of the proposed CRF model when using only word features *Fwc*.

| Type | P (%) | R (%) | F1 (%) |
|---|---|---|---|
| | 80.00 | 37.98 | 57.14 |

Secondly, we employed our proposed hybrid neural network and CRF models with added features X = $\{F_{wc}, F_{syllable}, F_{arg\_type}, F_{relation\_type}, F_{position}\}$, $F_{position, relation\_type, arg\_type}$, $F_{position, relation\_type, arg\_type, wc, position+relation\_type+arg\_type}$, and $F_{position, relation\_type, arg\_type, wc, position}$. The feature items that were selected for this paper are as follows: word feature/syllable/argument, type/relation, and type/position/syllable. The feature set was expressed as follows: X = $\{F_{wc}, F_{syllable}, F_{arg\_type}, F_{relation\_type}, F_{position}\}$, where the five elements were defined as follows.

1. *Fwc*: the word feature that represents the word itself.
2. $F_{syllable}$: the syllable that represents the suffix and prefix of the word.
3. $F_{arg\_type}$: the argument type, i.e., whether it is a first or second argument.
4. $F_{relation\_type}$: the relation type that represents the relation between two entities.
5. $F_{position}$: the entity position feature that represents the position of the word contained in each entity.

The relation extraction results indicated that the features of the words are vital, but the syllable directly reduces the F1 score of the relation extraction results. The Hyper parameters of the hybrid neural network and performance results are listed in Tables 15 and 16 respectively.

**Table 15.** Hyper parameters of the hybrid neural network.

| Parameter Description | Value |
|---|---|
| Dimension of word embedding | 300 |
| The number of hidden units in the encoding layer | 300 |
| The number of hidden units in decoding layer | 300 |
| Context window size of CNN module | 3 |
| The filter number of CNN | 100 |
| Dropout ratio | 0.3 |

**Table 16.** CRF model performed while using multiple feature comparisons with the neural network method on the task of relation extraction.

| Model | Type | P (%) | R (%) | F1 (%) |
|---|---|---|---|---|
| | $F_{wc}$ | 80.00 | 37.98 | 57.14 |
| | $F_{wc} + F_{relation\_type} + F_{position, relation\_type, arg\_type}$ | 85.00 | 42.34 | 61.34 |
| Our CRF method | $F_{wc} + F_{position} + F_{relation\_type} +$ $F_{position, relation\_type, arg\_type, wc, position+relation\_type+arg\_type}$ | 85.43 | 43.56 | 57.64 |
| | $F_{wc} + F_{position} + F_{relation\_type} +$ $F_{position, relation\_type, arg\_type, wc, position}$ | 84.77 | 41.94 | 56.11 |
| | $F_{wc} + F_{position} + F_{relation\_type} +$ $F_{position, relation\_type, arg\_type, wc, position+relation\_type+arg\_type}$ | 86.89 | 41.88 | 56.51 |
| Our hybrid neural network+Feature | $F_{position} + F_{relation\_type} + F_{arg\_type}$ | 65.2 | 38.1 | 48.7 |
| hybrid neural network [39] | - | 44.14 | 38.25 | 34.61 |
| CNN [1] | - | 57.4 | 25.6 | 35.4 |
| LSTM-LSTM [34] | - | 42.3 | 51.1 | 41.5 |

Finally, our proposed CRF model was applied to the same documents while using only $F_{wc}$ to determine the relation extraction performance for each of the five primary relation types individually.

It can be seen from the above results that the traditional and neural network methods have their own advantages and disadvantages. The CRF model is better than the neural network in relation extraction with Uyghur features. The F1 value of the hybrid neural network with these features was only 48.7, which was higher than that of the pure hybrid neural network method. Though this present work was hindered by the lack of relation extraction corpora in the Uyghur language for conducting comparison experiments, we conducted comparative experiments with our own corpus and different methods.

*4.3. Analysis and Discussion*

Finally, the training data obtained the following results under the condition of adding features in five different categories. The performance results for the individual relation types are listed in Table 17.

**Table 17.** Accuracy of relation extraction when using only word features $F_{wc}$ for specific relation types.

| Type | P | R | F1 |
| --- | --- | --- | --- |
| All | 83.33 | 26.32 | 40.00 |
| Aff.Employment | 75 | 33.33 | 46.15 |
| Aff.Owner | 100 | 50 | 66.67 |
| Social.Family | 75 | 33.33 | 46.15 |
| Social.Role | 100 | 20 | 33.33 |
| Whole.Geo | 66.67 | 42.34 | 51.78 |

We compared our method with different methods. The results in Table 17 are of particular interest in that they show the effect of the good relation extraction performance for Aff.Owner on the overall Uyghur relation extraction performance of the proposed method. The expanded corpus provides a higher quality basis for conducting Uyghur relation extraction research in NLP.

## 5. Conclusions

The present article addressed the many issues limiting Uyghur language research by proposing a hybrid neural network that combines a CRF model for making suggestions regarding annotation that is based on extracted relations with a set of rules applied by human annotators. Uyghur relation extraction was therefore implemented with the help of human correction to more rapidly expand the existing corpus than by human annotation alone. The relation extraction performance of our proposed approach was demonstrated by using an existing Uyghur corpus, and our method achieved a maximum F1 score of 61.34%.

In the future, we will explore how to better link these semantic features on the neural network. We also need to solve the problem of expanding the Uyghur relation extraction corpus and try to promote the recall value.

**Author Contributions:** A.H. and K.A. conceived and designed the experiments; A.H. performed the experiments; K.A. analyzed the data; T.Y. contributed materials; A.H. wrote the paper; T.Y. revised the manuscript. All authors have read and agreed to the published version of the manuscript.

**Funding:** This research was funded by the National Natural Science Foundation of China (Grant Nos. 61762084, 61662077, and 61462083), the Opening Foundation of the Key Laboratory of Xinjiang Uyghur Autonomous Region of China (Grant No. 2018D04019), and the Scientific Research Program of the State Language Commission of China (Grant No. ZDI135-54).

**Conflicts of Interest:** The authors declare no conflicts of interest.

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
