# Peer review of "Semi-Automatic Corpus Expansion and Extraction of Uyghur-Named Entities and Relations Based on a Hybrid Method"

_information, doi:10.3390/info11010031_

Round 1

Reviewer 1 Report

A presentation of the language under study, where it is spoken, by how many people and of its major characteristics is necessary for placing the experiment in a context and for understanding some details in the paper.

The first sentences/sentence in sections 1 and 2 must be removed. They seem to come from the template.

In section 2 the references between square brackets occur before the reference with names. They should be swapped.

Examples must be translated in English.

The type of annotation chosen must be provided an explanation.

Tables are not discussed.

Author Response

Dear Editor, 

We appreciate the constructive comments regarding our paper; our responses to your feedback follow.

   Please see the attachment.

Reviewer 2 Report

Major comments:

    1. Some of the related work is not cited, most notable of them is

Luo, Ling, Zhihao Yang, Pei Yang, Yin Zhang, Lei Wang, Hongfei Lin, and Jian Wang. "An attention-based BiLSTM-CRF approach to document-level chemical named entity recognition." Bioinformatics 34, no. 8 (2017): 1381-1388.       2. Classifier design is the exact one from Zheng, Suncong, Yuexing Hao, Dongyuan Lu, Hongyun Bao, Jiaming Xu, Hongwei Hao, and Bo Xu. "Joint entity and relation extraction based on a hybrid neural network." Neurocomputing 257 (2017): 59-66.       so there is no innovation there except for the corpus in Uyghur language. However, because this is not a paper about the this corpus, authors need to prove that their framework outperforms this approach taken as is. Namely, it is necessary to use their system as a baseline in your experiments. Without this comparison, innovation is questionable at best.

Minor comments:

Formulae are often not in line with the text (was the paper written in word?) - see, e.g. p.6 l.202 and so on - it needs to be fixed. p.2 l.52 - the claim that direct translation of knowledge graphs is not feasible should be supported by either citations or examples. p.3 l. 99 - examples given here are in Latin script, but what about your data? Is it in Arabic or in Latin script? This is not clear from the paper. p.3 l.110-111 - this work identifies entities, features are generated by ANN and never used outside the task, so the claim seems strange. p.4 l.126 - the middle word => do you assume that there is just one middle word? or 'a middle word'? it is not clear. p.4 Table 4 - formulae trouble again. p.6 l.190 - 'complicated feature engineering ' - this is a standard function of ANN, not more or less complicated than all other neural networks. Section 3.4 should be shortened dramatically because these standard NN architectures are not a contribution of this paper. Please just state staking order of your layers. p.9 l.291 - sentences filtered how? An explanation is needed. p.12 l.365 - name of one of the authors starts with lower case, please fix.

Author Response

(The authors gave the same response as above.)

Round 2

Reviewer 2 Report

My previous comments were adequately addressed.

In some places formulas are still not in line with the text.